# Percutaneous transhepatic drainage is safe and effective in biliary obstruction—A single-center experience of 599 patients

**Bálint Kokas[1], Attila Szijártó[1◉], Nelli Farkas[2], Miklós Ujváry[1], Szabolcs Móri[1], Adél Kalocsai[1], Ákos Szücs[1◉]***

**1** Department of Surgery, Transplantation and Gastroenterology, Semmelweis University, Budapest, Hungary, **2** Institute of Bioanalysis, University of Pécs, Pécs, Hungary

◉ These authors contributed equally to this work.
* szucs.akos@gmai.com

## Abstract

### Background

Historically, surgical bilioenteric bypass was the only treatment option for extrahepatic bile duct obstruction, but with technological advancements, percutaneous transhepatic drainage (PTD) and endoscopic solutions were introduced as a less invasive alternative. Endoscopic methods may lead to a decreasing indication of PTD in the future, but today it is still the standard treatment method, especially in hilar obstructions.

### Methods

In our retrospective data analysis, we assessed technical success rate, reintervention rate, morbidity, mortality, and the learning curve of patients treated with PTD over 12 years in a tertiary referral center.

### Results

599 patients were treated with 615 percutaneous interventions. 94.5% (566/599) technical success rate; 2.7% (16/599) reintervention rate were achieved. 111 minor and 22 major complications occurred including 1 case of death. In perihilar obstruction, cholangitis were significantly more frequent in cases where endoscopic retrograde cholangiopancreatography had also been performed prior to PTD compared to PTD alone, with 39 (18.2%) and 15 (10.5%) occurrences, respectively.

### Discussion

The results and especially the excellent success rates demonstrate that PTD is safe and effective, and it is appropriate for first choice in the treatment algorithm of perihilar stenosis. Ultimately, we concluded that PTD should be performed in experienced centers to achieve low mortality, morbidity, and high success rates.

**Funding:** The authors received no specific funding for this work.

**Competing interests:** The authors have declared that no competing interests exist.

## Introduction

Obstructive jaundice is a consequence of extrahepatic bile duct obstruction. Historically, surgical bilioenteric bypass was the only solution for this type of obstruction, but with technological advancements, percutaneous transhepatic drainage (PTD) and endoscopic retrograde cholangiography (ERCP) were introduced as less invasive alternatives [1]. Compared with ERCP, PTD is considered as more invasive, since in this intervention, the drain tube has to be punctured through the liver parenchyma, causing more tissue injury. Drain dislocation, drain-related pain or discomfort can affect the patients' quality of life, which is, of course, not present in endoscopic treatments [2, 3]. These adverse events contribute to that daily routine where ERCP and stent implantation is the first choice as a curative, bridging or palliative treatment for extrahepatic bile duct obstruction. However, several guidelines recommend PTD as the first choice of treatment, especially in hilar obstructions [4, 5]. Nevertheless, the type of intervention should be selected by multidisciplinary teams, based on the disease etiology and the localization of the obstruction. The continuous development of endoscopic ultrasound may lead to a decreasing indication of percutaneous drainage in the future, but today it is still widely accessible and is the standard of care [4], especially considering several scenarios (i.e. Roux-en-Y bilioenteric bypass, Billroth resection, etc.) where endoscopy would be difficult to perform. Adverse events related to percutaneous intervention can reach up to 61%, and mortality to 6% in some reports, but experienced centers can offer better result, for example Robson et al. reported 2% mortality, and Muller et al. reported 28% morbidity [6, 7]. The technical success of the percutaneous intervention is a major advantage, as it can be as high as 94–100% compared to less favorable results of ERCP [2, 8]. In our retrospective data analysis, we asses morbidity and mortality rates, and the learning curve of cases treated over 12 years in a tertiary referral center demonstrating excellent technical success and acceptable complication rates.

## Materials and methods

A retrospective data analysis was performed of the results of patients with biliary obstruction treated with PTD between 2007 and 2018 in our tertiary referral center. The study was approved by the Semmelweis University Regional And Institutional Committee of Science and Research Ethics (SE RKEB# 50/2021.). The patient records were accessed via the electronic medical system used by Semmelweis University. The records were fully anonymized at the data analysis, the ethics committee did not required for an informed consent.

### Patient population

615 percutaneous biliary interventions were performed in 599 patients with biliary obstruction. The interventions were performed with curative and palliative intention; preoperatively in 98 patients, followed by an operation with curative intention, postoperatively in 63 inoperable patients, and without operation in 436 upfront inoperable patients. The cause of biliary obstruction was diagnosed by radiologic findings (computer tomography, magnetic resonance imaging) and histology. The level of obstruction was determined via percutaneous transhepatic cholangiography (PTC). Perihilar obstructions were classified based on the Bismuth-Corlette (BC) classification. Distal bile duct obstruction was diagnosed between the cystic duct and the level of the pancreas; in case of the postoperative local recurrence of malignancy or anastomotic benign obstruction, no further localization was specified. Disease etiology and stenosis localization results are detailed in Table 1.

**Table 1. Patient population of 599 patients.**

| Patient population | n (total = 599) | % |
|---|---|---|
| Age (mean) | 65.1 (St.dev 12.07) | - |
| **Sex** | | |
| Male | 315 | 53% |
| Female | 284 | 47% |
| **Disease etiology** | | |
| Pancreatic head malignancy | 207 | 35% |
| Perihilar malignancy (Klatskin tumor) | 183 | 31% |
| Common bile duct malignancy | 20 | 3% |
| Vater papilla malignancy | 16 | 3% |
| Benign biliary stricture | 22 | 4% |
| Intrahepatic malignancy | 15 | 3% |
| Gall bladder malignancy | 44 | 7% |
| Other malignant disease | 92 | 15% |
| **Localization of the stenosis** | | |
| Benign anastomotic stricture (in a previous biliodigestive anastomosis) [*] | 5 | 1% |
| Intrahepatic | 14 | 2% |
| Local recurrence of malignancy [*] | 25 | 4% |
| Distal bile duct (below the cystic duct) | 198 | 33% |
| Common bile duct | 357 | 59% |
| Common bile duct stenosis: Bismuth Corlette Classification | | |
| *Common bile duct: Bismuth-Corlette type I* | *202* | *34%* |
| *Common bile duct: Bismuth-Corlette type II* | *21* | *4%* |
| *Common bile duct: Bismuth-Corlette type IIIa* | *20* | *3%* |
| *Common bile duct: Bismuth-Corlette type IIIb* | *24* | *4%* |
| *Common bile duct: Bismuth-Corlette type IV* | *90* | *43%* |
| **Serum bilirubin (mean)** | **mean: 365.4 (umol/l) St. dev. 193.6** | **median: 360 (umol/l)** |

[*] no further localization was defined.

n = number of patients.

## Technique

The interventions were performed by two physicians. All interventions were carried out with the patient in the supine position. Following premedication with the injection of promethazine (25-50mg), atropine (0.5-1mg), and pethidine (50-100mg), the patient was given local anesthesia consisting of 1% lidocaine at the puncture site. The primary puncture was performed in the 9th–10th intercostal space on the patient's right side. If left liver lobe decompression was indicated, left-side puncture was performed in the subxiphoid space. Once the Chiba needle (Cook Medical, Bloomington, IN, USA) was in the bile duct, as confirmed with cholangiography, a 0.018-inch wire (Cook Medical) was advanced and the needle was removed. A percutaneous access set (Cook Medical) with two sheaths and a metal cannula was used to introduce a cannula accepting a larger wire suitable for the planned intervention. After the coaxial tip was inserted into the bile duct using the 0.018-inch wire, the two inner components were removed, leaving the outer 4French (F) sheath behind. Cholangiography was performed to determine the obstruction level. A 4F biliary manipulation catheter (Cook Medical) was used to cross the obstructing lesion. The 0.018-inch wire was left in place to preserve the route for security

reasons. Finally, an 8.5F or 10.2F drain (Cook Medical) was left behind bridging the stricture. If crossing the obstruction was not possible, an external drain was left in place. The drain was sutured and fixed to the skin with its original kit.

## Variables and definitions

Technical success rate, reintervention rate, early complications and learning curves were assessed. Complications were further divided to minor and major groups. Corresponding to the Clavien-Dindo classification, the minor group matched grades I-II, and the major group matched grades III-V. Minor complications were observed, such as pancreatitis, cholangitis, bleeding, hepatic abscess, biloma and drain dislocation. Cholangitis was diagnosed if systemic inflammation, cholestasis, and biliary dilatation were present corresponding to the Tokyo Guidelines of diagnostic criteria of acute cholangitis [9]. The learning curves of the two physicians were assessed based on the internal-external to internal drainage ratio, with the assumption that high internal-external drainage percentage demonstrated the physician being more experienced. Learning curve was also assessed regarding the complication rates over the years.

## Statistical analysis

Descriptive statistics (number of events and percentage for categorical variables, and total number, mean, and standard deviation for continuous variables) were obtained using Prism Graphpad and the MS Excel 2016 software. To analyze correlation between categorical variables, Fisher-test (in case of a low number of events) and Chi-squared test were applied. All statistical calculations were done with the IBM-SPSS ver. 25 software (IBM SPSS Statistics for Windows, Version 25.0. Armonk, NY: IBM Corp.).

## Results

### Technical success and reintervention rate

The intervention was considered technically successful when internal or external drain could be left in the bile ducts and the final PTC confirmed adequate biliary drainage. Thus, technical success was achieved in 94.5% (566/599) of the patients. In 33 cases, it was technically pointless to drain, as the subsegmental obstruction resulting from the advanced state of the disease could not be resolved with even multiple drainages. The technical success rate was measured in the perihilar subgroup as well: 96.3% (344/357). 16 patients needed reintervention in 30 days due to complications like dislocation (n = 13), drain obliteration (n = 2), or haemobilia (n = 1). The calculated 30-day reintervention rate is 2.7% (16/599).

### Complications

Intervention-related early complications were divided into minor and major groups (Table 2). Minor complications were the following: Pancreatitis–defined by the Revised Atlanta Classification–was observed in 7 patients. Bleeding was registered as a complication if red blood cell transfusion was needed. With that consideration, 5 cases of bleeding were registered. Biloma and hepatic abscess were observed in 4 cases, which needed no surgical intervention. 71 patients with cholangitis were registered after the intervention. 63 drain dislocations were noted. Out of these 63 complications, 39 happened within 30 days after the intervention (early dislocation) and 24 happened later than that (late dislocation). From the 39 early dislocation 24 were managed without intervention. In these cases there were different disease courses that did not indicate a reintervention. Such case scenarios were: reposition was managed without

**Table 2. Minor and major complications after percutaneous intervention in 599 patients.**

| Minor complications (Clavien Dindo I-II) | n | % |
|---|---|---|
| Bleeding (transfusion needed) | 5 | 0.01 |
| Cholangitis (after the PTD) | 71 | 12 |
| Early dislocation (managed without intervention) | 24 | 4 |
| Pancreatitis | 7 | 1 |
| Intraabdominal abscess | 3 | 0.5 |
| Intraabdominal biloma | 1 | 0.17 |
| **Total** | **111** | **19** |
| Major complications (Clavien Dindo III-V) | n | % |
| Perforation | 1 | 0.17 |
| Bleeding (2 managed with reoperation, 1 with reintervention) | 3 | 0.5 |
| Pancreatitis | 1 | 0.17 |
| Early dislocation and obliteration (managed with reintervention) | 15 | 3 |
| Intraabdominal abscess | 1 | 0.17 |
| Death | 1 | 0.17 |
| **Total** | **22** | **4** |

n = number of patients.

true radiological intervention, internal-external drain dislocated to external position, resolution of cholangitis and dilated bile ducts, disease progression or other organ failure.

Major complications were observed in 22 cases. In 1 case, bile duct perforation with intraabdominal drain dislocation and bile leakage resulted in peritonitis and required operation. In 4 cases, other etiologies (pancreatitis with necrosis, hepatic abscess, and intraabdominal bleeding in two cases) required surgical laparotomy, in 15 cases drain dislocation or obliteration, and 1 case haemobilia was managed with reintervention.

In 1 case, PTD-related small bowel perforation resulted in biliary peritonitis, reoperation and death. The intervention related mortality is 0.17% (1/599).

## Complications in the perihilar obstruction subgroup

In this subgroup, we assessed the obstructions classified by BC, mentioned in Table 1. In total, 357 perihilar obstructions were found. Percutaneous intervention was performed after failed ERCP in 214 patients, and 143 PTD without ERCP were done in the mentioned subgroup. Failed ERCP included unsuccessful stent implantation or ineffective drainage. All endoscopic interventions were performed in other institutes. Complication data of percutaneous intervention alone (hilar PTD) and ERCP followed by percutaneous intervention (hilar ERCP+PTD) were analyzed and compared in the hilar obstruction subgroup. 30 minor and 8 major complications were observed in the hilar PTD group, and 39 minor and 12 major complications were found in the hilar ERCP+PTD group (Table 3). The difference between the two groups (hilar ERCP+PTD vs hilar PTD) was statistically not significant (p = 0.557) regarding the total number of complications. We analyzed cholangitis separately as well, as it was the complication with the highest numbers in both subgroups. The other complications were not compared statistically by type because of the low number of events. In the hilar PTD group, 15 cases of cholangitis were observed before and 23 after the intervention. Nevertheless, in the hilar ERCP+PTD group, 39 cases were observed before the percutaneous drainage (between the ERCP and PTD), and 25 after the percutaneous drainage. The number of cholangitis observed before the intervention in the hilar ERCP+PTD subgroup was significantly higher than the other

**Table 3. Minor and major complications in the hilar PTD group of 143 patients, and the hilar ERCP+PTD group of 214 patients.**

| Minor complications (Clavien-Dindo I-II) | n (%)–hilar ERCP+PTD | n (%)–hilar PTD group | p |
|---|---|---|---|
| Bleeding | 4 (2) | 0 | |
| Biloma | 1 (0.5) | 0 | |
| Intraabdominal abscess | 1 (0.5) | 1 (0.7) | |
| Cholangitis (after the PTD) | 25 (12) | 23 (16) | 0.232 |
| Early dislocation (managed without intervention) | 8 (4) | 1 (0.7) | |
| Pancreatitis | 0 | 5 (3.5) | |
| Total | 39 (18) | 30 (21) | 0.518 |
| **Major complications (Clavien-Dindo III-V)** | **n (%)–hilar ERCP+PTD group** | **n (%)–hilar PTD group** | **p** |
| Perforation | 1 (0.5) | 0 | |
| Bleeding (managed with reintervention or reoperation) | 2 (0.1) | 0 | |
| Pancreatitis | 0 | 1 (0.7) | |
| Early dislocation and obliteration (managed with reintervention) | 9 (4) | 6 (4) | |
| Intraabdominal abscess | 0 | 1 (0.7) | |
| Total | 12 (6) | 8 (6) | 0.996 |
| **Major and minor complications** | **n (%)–hilar ERCP+PTD group** | **n (%)–hilar PTD group** | **p** |
| **Total** | **51 (24)** | **38 (27)** | **0.557** |

n = number of patients.

group (p = 0.046), (Fig 1). The numbers of cholangitis observed after the intervention were not different statistically in the subgroups (p = 0.232).

## Improving results

All interventions were performed by two physicians who were getting more and more experienced through the years. This allowed us to try to measure their improvement with a learning curve. We analyzed the success rates of the percutaneous interventions on an annual basis. External decompression and internal-external decompression were differentiated and counted for every single year in the investigated period. Internal-external decompression was considered as the best experience, being superior to the external method. An annual increase was observed in the number of the internal-external drainage as a result of the physicians getting more and more experienced (Fig 2A). After 203 PTDs, the rate of internal-external/external drainage ratio stabilized above 1.

Yearly complication rates were also calculated. A higher peak point has been found during the early years and a decreasing complication rate over the years. This peak corresponds to the early courage of the less experienced interventionists which decreased with the rising experience (Fig 2B). Although this does not follow precisely the learning curve on the other figure but, a major decrease can be observed in the number of complications, approximately between year 2011–2012 (X-axis) where the internal/external ratio stabilized above 1.

## Discussion

Obstructive jaundice increases the risk of morbidity and mortality through several pathophysiologic changes [10]. Percutaneous transhepatic biliary drainage is a widely used interventional

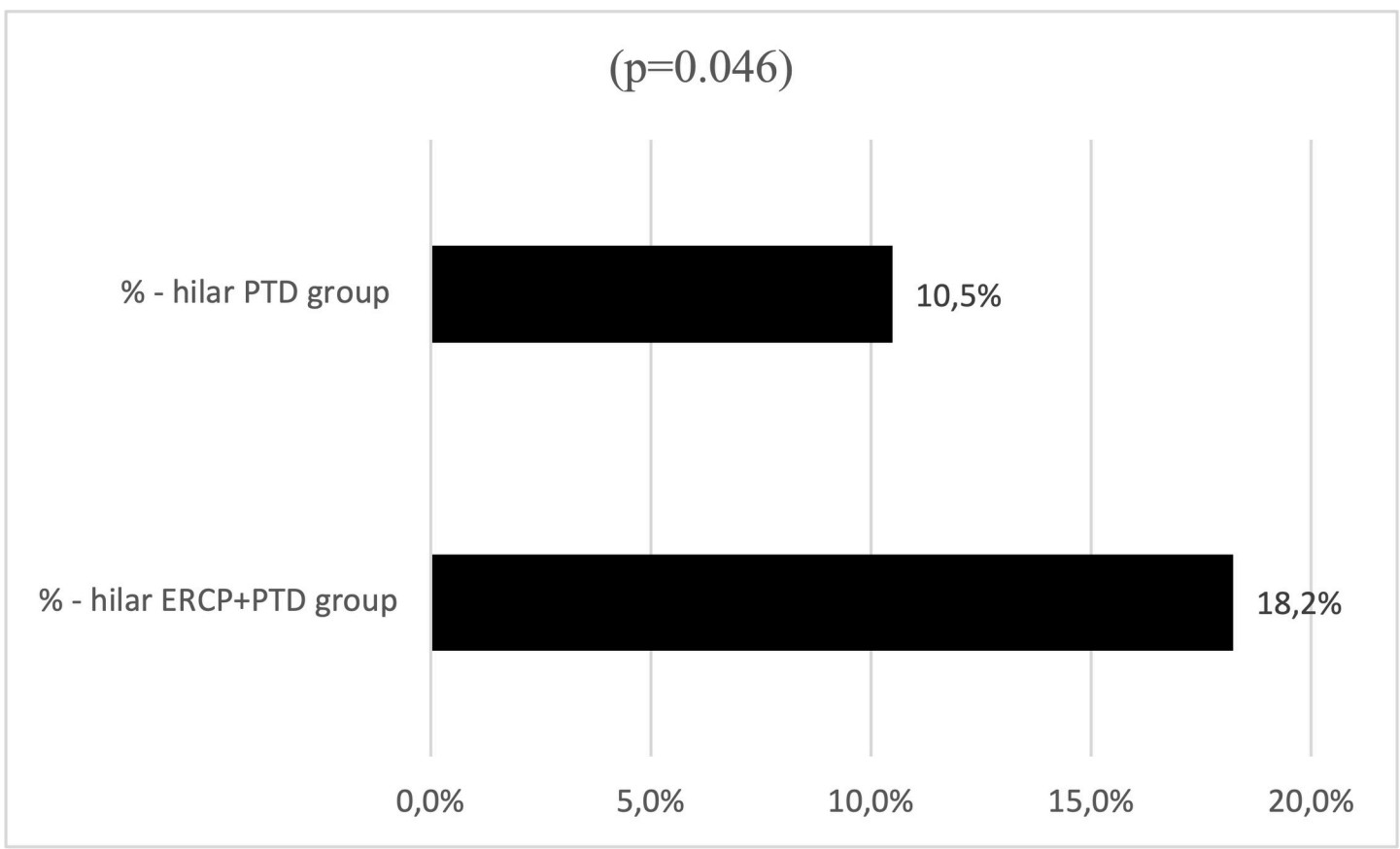

**Fig 1. Cholangitis in the hilar PTD group of 143 patients and the hilar ERCP+PTD group of 214 patients before the PTD intervention.** Statistically significant difference. p = 0.046.

method to relieve mechanical jaundice regardless of the localization of the obstruction. In this study, we retrospectively analyzed 599 patients with mechanical biliary obstruction receiving percutaneous biliary drain. To the best of our knowledge, this study has the largest patient population in a single center unit in this setting. The high bilirubin levels in our patient population (median serum bilirubin 360 umol/L) indicates the advanced stage of their disease and the low compensatory capability, similar to the study populations in papers published by Sut el al. (median serum bilirubin 397 umol/L) or Robson et al. (median serum bilirubin 201 umol/L) [2, 11].

Despite the low performance status of our study group, we could achieve a high technical success rate (94.5%). In some cases, subsegmental obstructions cause noticeable hyperbilirubinemia, however, these bile ducts are not suitable for percutaneous drainage. Other publications about PTD like Robson et al. published 100% technical success, Garcarek et al. reported 96.2%, Kloek et al. reported 100%. Also, Kloek et al. in the same paper report 81% in endoscopic biliary drainage groups. Walter et al. reported 98% technical success rate after PTD and 78% after endoscopic drainage in 129 patients with Klatskin tumor treated in their study [12]. Vitte et al. published 76% and 86,9% of success in low and high volume endoscopic centers respectively [13]. However, the meta-analysis including the previously mentioned study reports different endoscopic success rates (76–99%) as a result of heterogenous definitions of procedural success [14]. The meta-analysis of Zhao et al. analyzing five retrospective and three randomized controlled trials with a total of 692 participants also reported a trend towards the

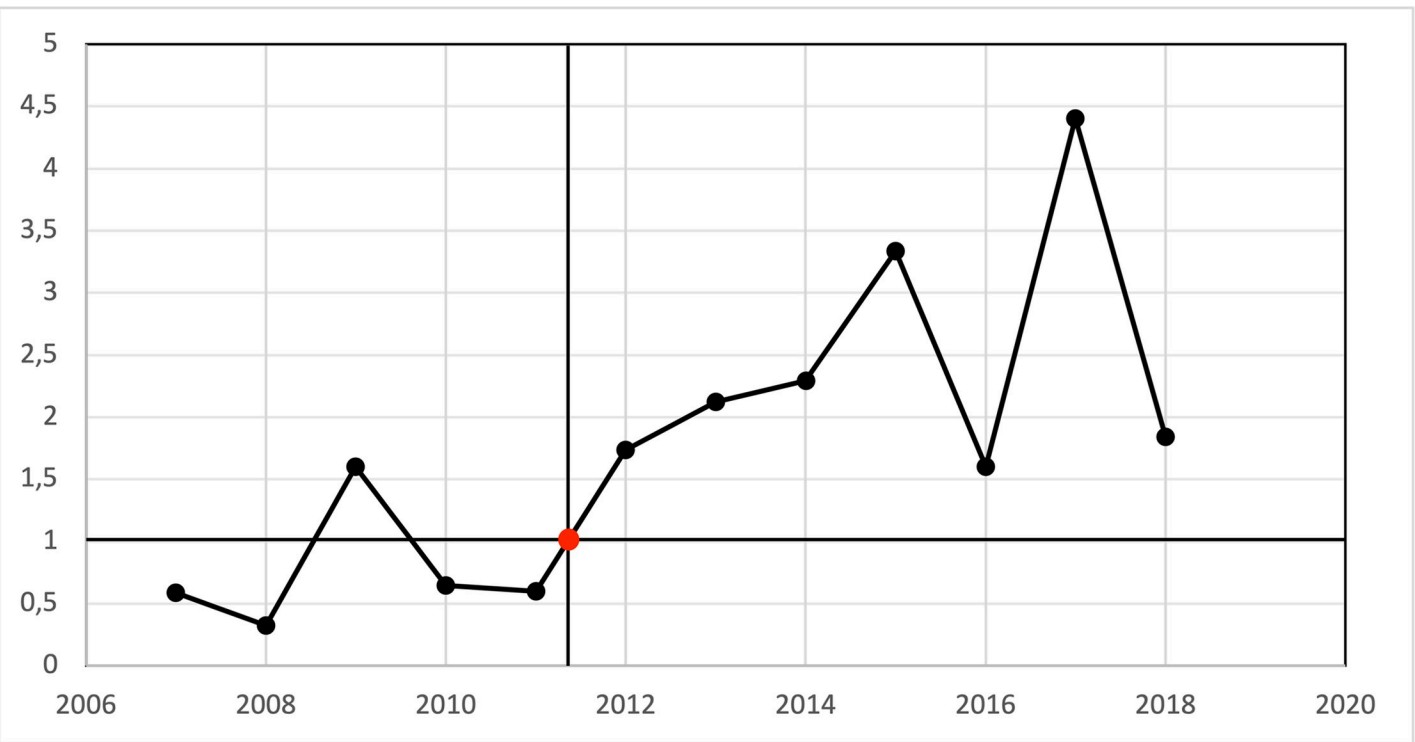

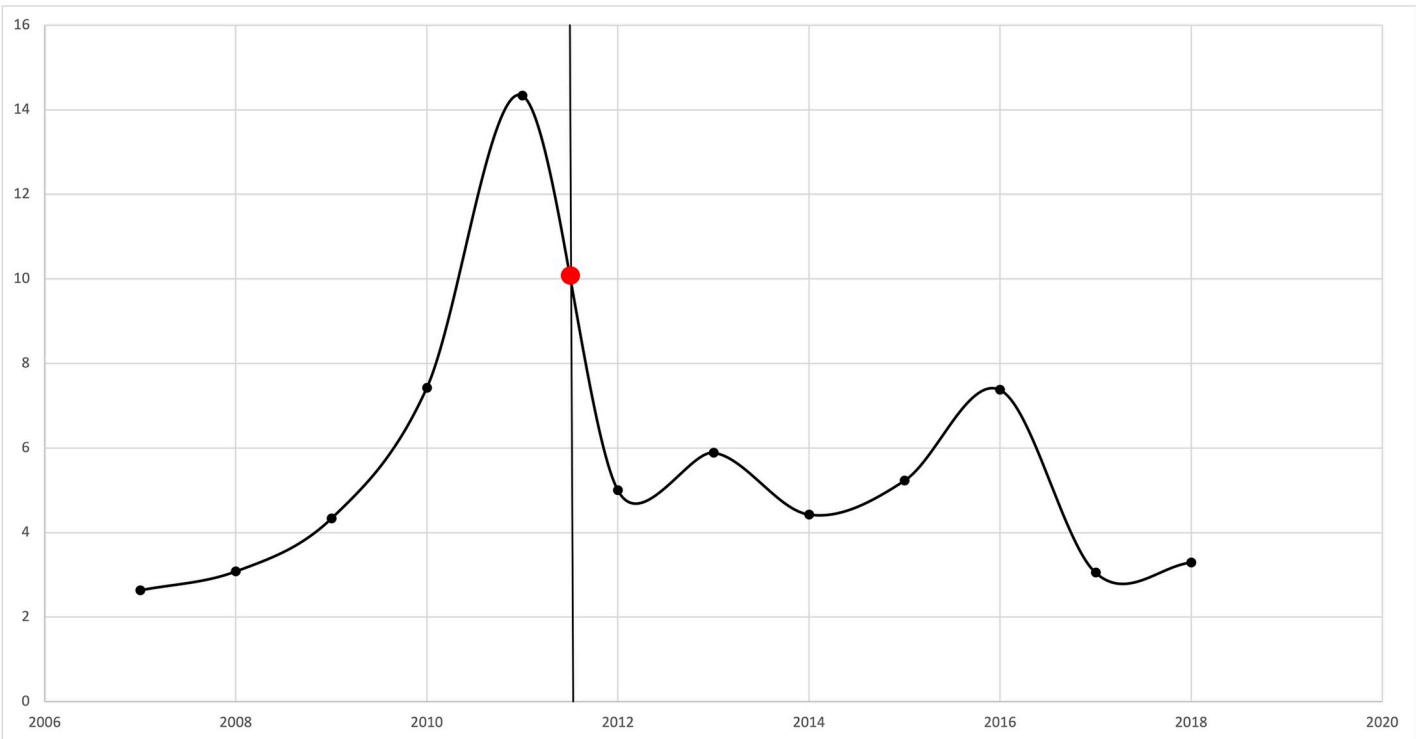

**Fig 2.** A—Learning tendency. X axis—years, Y axis—internal-external/external drainage ratio X axis–years. The red dot shows the point where the ratio stabilized above 1 (203 PTDs). B–Learning tendency. X axis–years, Y axis–total number of interventions / total number of complications ratio. The red dot shows the point where between 2011–2012 the internal-external/external ratio stabilized above 1 and the complications started to decrease.

advantages of PTD over endoscopic drainage from the aspect of technical success (odds ratio (OR), 2.18; 95% CI, 0.73–6.47; P = 0.16). Another meta-analysis of Tang et al. comparing the two biliary drainage methods in 1030 patients found that PTD was associated with a higher successful rates of palliative relief of cholestasis (RR = 1.20, 95% CI: 1.10–1.31; P < .0001) [2, 8, 15–18].

The primary goal in biliary obstruction is the resolution of the obstruction and the adequate biliary drainage to relieve imminent cholangitis. Considering that in the majority of the cases PTD happens in an advanced state of the disease, or as a salvage therapy clinical success is hard to interpret. That is why rather technical success was investigated in this study. However, there are several other factors that play an important role in decision making about the technique used to solve the obstruction. Disease etiology, clinical success, quality of life are important in the long term success. However, the literature is not always clear about the best treatment in different disease etiology. The meta-analysis of Huszar et al. found no significant difference between endoscopic, percutaneous or surgical interventions in benign biliary obstructions regardless of the localization, Saluja et al. also found no significant difference between the endoscopic and percutaneous groups in malignant obstructions in the long term patency rates [19]. Quality of life and patient preference are also important knowing that percutaneous drain often needs maintenance, emptying or flushing or internalization of the drain and a painful drain can cause bad patient compliance, besides external drainage results in loss of bile [2, 3, 20]. Not to mention puncture site seeding metastasis which can be also problematic. These factors have to be well considered before making long term therapeutical decisions. Furthermore, complications also play an important role in therapeutical decisions.

In our study population only one patient suffered a lethal complication (0.17%), and only 4% (22/599) of the cases were diagnosed with major complications, which is favorable. After percutaneous interventions, Robson et al. reported 2%, while Mueller et al. reported 0.015% of the interventions resulting in death. Andriulli et al. found 0,33% mortality after ERCP in their systematic review survey including 21 prospective studies and 16.855 patients [2, 21].

Intervention related bowel perforations are major but rare complications thus few studies mention this event. Stapfer et al. and Howard et al. report 0.6–1% of duodenal perforations after ERCP [22, 23]. Only 1 perforation was observed in 261 patient in the largest study reporting perforation after percutaneous intervention [24]. We also had only 1 (0.17%) of this adverse event.

We found 1 severe post PTD pancreatitis (0.17%) in our study group. Andriulli et al. found 3.47% of post ERCP pancreatitis in 16.855 patients in their study. Pancreatic damage was mild in 44.8% in 43.8%; severe in 11.4% of all patients with post-ERCP pancreatitis, or 0.40% of all investigated patients [20]. A study comparing ultrasound and fluoroscopy guided PTD in 195 patients reported no pancreatitis after the PTD [25]. Another meta-analysis about preoperative biliary drainage in hilar cholangiocarcinoma showed that the total incidences of pancreatitis in the endoscopic biliary drainage group was 11.9% (21/157), in contrast to none in the PTD group [26]. This shows a clear advantage of PTD over endoscopic method regarding post-interventional pancreatitis.

The incidence of post-interventional bleeding ranges widely because of differences in definition. In a study conducted by Freeman et al. reported clinically significant hemorrhage in 48 patients (2%) that was, moderate (up to 4 units of blood transfusion were needed) in 22 (0.9%), and severe (i.e., it necessitated the transfusion of 5 or more units of blood, surgery, or angiography) in 12 (0.5%), [27]. In the study already mentioned of Andriulli et. al, bleeding occurred in 226 patients (1.34%, CI 1.16–1.52%); it was moderate in the majority of cases (70.8%), severe in 66 cases, and associated with death in 8 patients; the bleeding-related mortality rate was 3.54% (CI 1.08–6.00%), [20]. Bleeding after PTD is also a well-known

complication, given the anatomic location of the intraparenchymal vessels. Nennstiel et al. described 7.7% of bleeding complications after PTD of which only 1 of 252 patients was considered major adverse event [25]. Rivera-Sanfeliz et al. reported higher incidence (8/346) requiring intervention to cease the bleeding [28]. In our presented data we observed 2 major bleedings.

Drain dislocation is a common adverse event as it appears from our data. In Tables 2 and 3 it stands out that it has the second highest number among the major and minor adverse events. ERCP stent migration is a similarly feared complication which can lead to bowel perforation as well. In a meta-analysis comparing percutaneous and endoscopic biliary drainage the total incidence rates of dislocation were 7.7% (12/156), and 18.1% (32/177) respectively. Our combined dislocation rate was 11% (63/599).

Based on these publications there is a slight advantage of percutaneous approach over ERCP regarding the above-mentioned major complications. Although complication data is hard to compare due to patient heterogeneity and the different methodologies applied.

In our study group, 19% (111/599) minor adverse events were observed.

Taking a closer look at our data, we found that in 11%, (63/599) dislocation and in 12%, (71/599) cholangitis is the observed adverse event. Bleeding, pancreatitis, abscess and biloma only occurred in a minority of the cases 3% (16/599).

Our data demonstrates that cholangitis and drain dislodgement are the two most common adverse events after percutaneous drainage, which is in accordance with the data published by Nennstiel et al. and Asadi et al. [7, 29].

Dislocated drain can be the result of bad patient compliance caused by low performance status or pain occurring after the intervention [2, 3]. Nevertheless, percutaneous drainage is not the only factor behind cholangitis. The previously mentioned meta-analysis of Zhao et al. comparing five retrospective studies and three randomized controlled trials reported significantly higher rates of biliary infection after endoscopic intervention than after PTD (OR, 0.59; 95% CI, 0.37–0.93; P = .02) [8].

A large proportion (60%) of our patients received PTD only after failed endoscopy. This reflects current practice, since most patients are referred to PTD providing centers with endoscopic stent in situ or after failed ERCP [30, 31]. High pre-interventional cholangitis rates also indicate the supposed adverse event of previous endoscopic intention and raises the question whether ERCP or PTD is the first choice of treatment in obstructive jaundice. Authors attempt to answer this highly debated question in several meta-analyses [32, 33]. The level of bile duct obstruction predicts the suitability of the endoscopic or the percutaneous method. In distal or BC type I-II obstruction, endoscopic intervention is less technically demanding, and is often the first choice in perihilar obstruction. However, in more advanced BC type III-IV obstructions, PTD is more beneficial, as it is less likely to causes cholangitis, as discussed in several guidelines [4, 5, 34]. Therefore, we analyzed the perihilar subgroup in our patient population. The difference between the hilar PTD and hilar ERCP+PTD groups from the perspective of complications was minimal. But only looking at the cholangitis numbers before the percutaneous intervention, we noticed a statistically higher number in the hilar ERCP+PTD group. This suggests that the patients who underwent ERCP before PTD had a higher chance of biliary infection. As the endoscopic procedures were done in other hospitals, during the retrospective data collection data availability was limited regarding the number of patients suffering from cholangitis before the ERCP. Although we do not have the data of patients who successfully underwent ERCP without cholangitis, this result still suggests that in perihilar obstruction, percutaneous access could be less harmful. This suggestion supports the recommendations of the above-mentioned guidelines. Presumably, further randomized controlled trials could clarify the beforementioned suggestions.

Speaking of failed ERCP, it has to be noted that the obstruction which sustain cholangitis was not solved by ERCP and another intervention was urgently needed to save the patient. Better patient selection could improve the rate of failed ERCP-s, indicate adequate primary PTD and solve the obstruction and the cholangitis by itself, or with further conservative treatment.

After failed ERCP, endoscopic-ultrasound-guided biliary drainage (EUSBD) also has to be mentioned. Other authors and guidelines suggest that EUSBD should be preferred over PTD after failed ERCP [35]. However, the benefit of this procedure over the percutaneous intervention treating malignant biliary obstruction is still not clear [36]. The technically demanding EUSBD is still lagging behind the PTD in accessibility, thus PTD remains standard of care in most medical centers [37]. In our patient population, there were no cases with endoscopic-ultrasound-guided biliary drainage prior to PTD.

The most suitable method for the resolution of benign biliary strictures is debated in a meta-analysis published by Huszar et al. [38]. In their data, percutaneous intervention was only superior to using endoscopic single plastic stent, and it is still not clear which is the best minimally invasive technique for the treatment of this etiology. When the obstruction cannot be solved with non-surgical and surgical methods, percutaneous transhepatic drainage combined with corticosteroid injection and balloon dilatation could be an effective treatment choice [39].

In perihilar stenosis, the invasion of the hepatic duct can be crucial from the perspective of operability. This factor is important in surgical planning, especially when liver resection is expected, and the future liver remnant requires preoperative biliary drainage [40, 41]. In this setting, the percutaneous approach is the most effective method for preoperative diagnosis, planning and drainage [39]. PTD is not only beneficial preoperatively, but is also safe and effective intraoperatively in liver and bile duct resections, preventing complications and anastomotic bile leaks [42].

Learning curves are good indicators of gaining experience and improving results [43]. We investigated the rate of internal-external/external drainages believing that internal-external drainage is the better outcome. Higher ratio was considered better. After 203 percutaneous drainages, the ratio stabilized above 1, which translates to the higher rate of internal-external drains. Still, a curve could not be drawn the improvement tendency is clear.

The main limitation of the study is retrospective data collection, the patient composition regarding the many cases with failed ERCP, and the limited data available on the previous endoscopic intervention.

## Conclusion

The results and especially the excellent success rates demonstrate that PTD is safe and effective, and it is appropriate for first choice in the treatment algorithm of perihilar stenosis. Ultimately, we concluded that PTD should be performed in experienced centers to achieve low mortality, morbidity, and high success rates.

## Supporting information

**S1 Table. Raw dataset.**
(XLSX)

## Author Contributions

**Conceptualization:** Bálint Kokas, Attila Szijártó, Ákos Szücs.

**Data curation:** Bálint Kokas, Miklós Ujváry, Szabolcs Móri, Adél Kalocsai.

**Formal analysis:** Bálint Kokas, Nelli Farkas.

**Investigation:** Bálint Kokas, Attila Szijártó, Ákos Szücs.

**Methodology:** Bálint Kokas, Attila Szijártó, Nelli Farkas, Ákos Szücs.

**Project administration:** Attila Szijártó, Ákos Szücs.

**Supervision:** Attila Szijártó, Ákos Szücs.

**Validation:** Attila Szijártó, Ákos Szücs.

**Writing – original draft:** Bálint Kokas.

**Writing – review & editing:** Bálint Kokas, Attila Szijártó, Ákos Szücs.

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
