## [Decision Letter · Decision Letter 0]

9 Jul 2021

PONE-D-21-17286

Percutaneous transhepatic drainage is essential in perihilar biliary obstruction – A single-center experience of 599 patients

PLOS ONE

Dear Dr. Szücs,

Thank you for submitting your manuscript to PLOS ONE. After careful consideration, we feel that it has merit but does not fully meet PLOS ONE’s publication criteria as it currently stands. Therefore, we invite you to submit a revised version of the manuscript that addresses the points raised during the review process.

We look forward to receiving your revised manuscript.

Kind regards,

Leonidas G Koniaris, MD

Academic Editor

PLOS ONE

Additional Editor Comments:

Please address reviwer comments.

Journal Requirements:

2. In ethics statement in the manuscript and in the online submission form, please provide additional information about the patient records used in your retrospective study. Specifically, please ensure that you have discussed whether all data were fully anonymized before you accessed them and/or whether the IRB or ethics committee waived the requirement for informed consent. If patients provided informed written consent to have data from their medical records used in research, please include this information.

" The funders had no role in study design, data collection and analysis, decision to publish, or preparation of the manuscript."

Reviewers' comments:

Reviewer's Responses to Questions

**Comments to the Author**

1. Is the manuscript technically sound, and do the data support the conclusions?

Reviewer #1: Partly

Reviewer #2: Partly

2. Has the statistical analysis been performed appropriately and rigorously? 

Reviewer #1: Yes

Reviewer #2: Yes

3. Have the authors made all data underlying the findings in their manuscript fully available?

Reviewer #1: Yes

Reviewer #2: No

4. Is the manuscript presented in an intelligible fashion and written in standard English?

Reviewer #1: Yes

Reviewer #2: Yes

5. Review Comments to the Author

Reviewer #1: This is a retrospective analysis of 599 patients who underwent percutaneous transhepatic drainage, evaluating success rate, morbidity and mortality, and learning curve over 12 years.

Table 1 lists the demographics, disease type, and location of lesion. Table 2 lists the major and minor complications. Table 3 lists the complications in the hilar PTD and PTD + ERCP groups. Figure 1 graphs cholangitis in the hilar PTD and ERCP + PTD groups. Figure 2 graphs the learning curve of the interventionalists.

Major Issues:

All ERCPs were done at outside institutions, and so only compares PTD to failed ERCPs. To claim PTD is essential, this would require a comparison to successful ERCP as well.

For Figure 1: From the text it is not clear regarding the timing of the of cholangitis. Is this saying that there is a higher rate of cholangitis after the ERCP, but before the PTD? Or is the cholangitis before both the ERCP and PTD? If you are saying that the cholangitis is after the ERCP, but before the PTD, it is important to state how many of the patients also had cholangitis prior to the ERCP.

I think your conclusions are overstated. You did not compare PTD to ERCP, but compared it to itself and failed ERCP. I think at most you can say that it is safe and effective and is appropriate for first choice in the treatment algorithm. Please consider revising this as well as the title of the manuscript.

Minor Issues:

The formatting of Table 1 can be improved specifically where there are subcategories or lab values.

Table 3, please include p values for the totals that are mentioned in the text.

Please explain more about the complication rate per year as compared to the rate of internal drainage experience curve.

Reviewer #2: Thank you for the opportunity to review this manuscript by Kokas et al. titled “Percutaneous transhepatic drainage is essential in perihilar biliary obstruction – A single center experience of 599 patients.” In this study, the authors retrospectively evaluate their experience performing 615 PTDs in 599 patients over a 12-year period. The number of patients included in this study is impressive. The authors report a technical success rate of 94.5%. Periprocedural mortality occurred in one patient, minor complications in 126 patients (21%), and major complications in 6 patients (1%). Among the 357 patients that underwent PTD for perihiliar biliary obstruction, minor complications developed in 27 patients and major complications in 4 patients. The authors conclude that PTD should be the first choice in the treatment algorithm of perihilar stenosis.

The number of patients included in this manuscript represents the main novelty of the paper; however, the remaining data reported are similar to a number of previously published manuscripts (retrospective studies, randomized controlled trials, systematic reviews and meta-analyses). Additionally, the discussion reads as an editorial with the opinions of the authors and lacks detailed discussion of the literature in context to the current study. I have clarified this further with several comments outlined below:

Major

1. Discussion – the authors advocate for PTD over ERCP; however, the current study does not compare PTD to ERCP. Additionally, the discussion fails to discuss/compare complications after PTD in the context of previously published literature regarding outcomes after ERCP in the setting of biliary obstruction. The discussion broadly approaches this but fails to discuss/compare the two in any granularity. As written, the current manuscript highlights that PTD can be performed safely in biliary obstruction with experienced clinicians.

2. Discussion/Conclusion – The authors conclude that complications rates after PTD are comparable to ERCP and subsequently the authors advocate for PTD as the first choice for perihilar biliary obstruction. (1) The authors study does not compare complications rates between PTD and ERCP. (2) The discussion does not discuss complication rates/types/severity after ERCP in the context of previously published literature. As written, how can the authors make these conclusions?

3. Results – What was the technical success rate among the subgroup of patients that underwent PTD for perihilar biliary obstruction? This appears to be the key point of the manuscript, yet the technical success rate among this subgroup is not reported. As the authors point out, this represents a more technically challenging procedure.

4. Title – “Percutaneous transhepatic drainage is essential in perihilar biliary obstruction” is a strong statement. The title should be revised to avoid “overselling” the study, as this is a retrospective observational report and not a randomized controlled trial (this study does not compare primary PTD to primary ERCP). Additionally, the number of patients listed in the title is not actually the number of patients with perihilar biliary obstruction and should be revised or changed (357 patients had perihilar biliary obstruction).

5. Results – the authors report 63 events where drain dislodgement occurred – these were classified as Clavien-Dindo Grade I or II complications (“minor” by the authors definition). Did none of these patients required a repeat intervention to have the drains replaced? By the authors definition, repeat intervention would be considered a “major” complication (Clavien-Dindo Grade III complications). Can the authors comment on how these 63 patients were managed if they did not require drain replacement? In a previous subsection, the authors state that 16 patients required reintervention in a 30-day period – were these counted as Clavien-Dindo III+ complications? It does not appear they were.

6. Discussion – Should technical success rate be the primary metric in determining approach to biliary drainage for biliary obstruction? Several other factors play an important role in this decision making (i.e., therapeutic success rate, quality of life, durability, etiology/pathology, etc). Can the authors comment on this in their discussion?

7. Results – cholangitis was not included as a “minor” complication after PTD for perihilar biliary obstruction – what is the rationale for excluding this complication, as it clearly his clinical implications?

Minor

1. Abbreviations (manuscript) – All abbreviations should be introduced and written out at first mention, followed by the abbreviated form for the rest of the manuscript.

2. Table 1 – what does the “*” denote?

3. Results – The first paragraph discussing one mortality should be included in the “complications” subsection and not as the first paragraph of the results section.

4. Results (page 9, line 140-141) – The definition of Clavien-Dindo “minor” complication does not need repeating in the results section, as it is defined in the methods.

5. Discussion – Paragraph 1 (a two-sentence paragraph) should be combined with paragraph 2

6. Discussion – throughout the manuscript the authors use the term “percutaneous transhepatic drain” but change to “percutaneous biliary catheter” in the discussion – terminology should remain consistent throughout the manuscript.

7. Discussion (paragraph 2) – the authors introduce new data from their study in the discussion section (median serum bilirubin concentration) – all data should be included in the results section of the manuscript.

8. Results – Procedural mortality was 1/599 – the authors report this as 0.001% - this is actually 0.17%.

6. PLOS authors have the option to publish the peer review history of their article (what does this mean?). If published, this will include your full peer review and any attached files.

Reviewer #1: **Yes: **Daniel Milgrom

Reviewer #2: No

---

## [Author Response · Author response to Decision Letter 0]

30 Oct 2021

Leonidas G Koniari, MD

Academic Editor

PLOS ONE

Submission Date: 11 august, 2021

Rebuttal letter

PONE-D-21-17286

Percutaneous transhepatic drainage is essential in perihilar biliary obstruction – A single-center experience of 599 patients

Thank you for considering our manuscript for publication. We also appreciate the time and effort you and each of the reviewers have dedicated to providing insightful feedback on ways to strengthen our paper. We have incorporated changes that reflect the detailed suggestions you have graciously provided. We also hope that our edits and the responses we provide below satisfactorily address all the issues and concerns you and the reviewers have noted.

The following document is a point-by-point response to the questions raised by the academic editor, Reviewer #1 and Reviewer #2, respectively. The marked-up manuscript with the changes highlighted will be uploaded separately with the requested raw dataset containing the minimum level of detail necessary to reproduce all numbers reported in the manuscript. We have updated the tables and figures as well, and included one new figure (Fig2. B).

Dear Dr Koniari,

Thank you for considering our manuscript for publication. We understand the questions raised by you and with your help we have improved the quality of the article.

Issue #1: “Please ensure that your manuscript meets PLOS ONE's style requirements, including those for file naming. The PLOS ONE style templates can be found at …"

Response: Thank you for the comment. We followed the information provided on the links and revised the styling in order to meet the PLOS ONE style requirements. 

Issue # 2: In ethics statement in the manuscript and in the online submission form, please provide additional information about the patient records used in your retrospective study. Specifically, please ensure that you have discussed whether all data were fully anonymized before you accessed them and/or whether the IRB or ethics committee waived the requirement for informed consent. If patients provided informed written consent to have data from their medical records used in research, please include this information.

Response: We understand the request. The patient record were accessed via the electronical medical system used by the Semmelweis University. The records were fully anonymized at the data analysis, the ethics committee did not required for an informed consent. The anonymized dataset will be available as a supporting information. As most of the patient of the presented study have been already passed away consent for publication of raw data not obtained but dataset is fully anonymous in a manner that can easily be verified by any user of the dataset. Publication of the dataset clearly and obviously presents minimal risk to confidentiality of study participants. It includes no direct identifiers and fewer than three indirect identifiers. 

Semmelweis University Regional and institutional committee of science and research ethics contact information:

1091 Budapest, Hungary

Üllői út 93. fsz. 2.

Telefon: 215-7300/53513 

Fax: 215-6228/ 53513

e-mail: titkarsag.kutatasetikai-bizottsag@semmelweis-univ.hu

Issue #3: Thank you for stating the following financial disclosure:

" The funders had no role in study design, data collection and analysis, decision to publish, or preparation of the manuscript."

a. Please clarify the sources of funding (financial or material support) for your study. List the grants or organizations that supported your study, including funding received from your institution.

d. If you did not receive any funding for this study, please state: “The authors received no specific funding for this work.”

Response: The statement - " The funders had no role in study design, data collection and analysis, decision to publish, or preparation of the manuscript." – was misinterpreted by the authors. The authors received no specific funding for this work, thus this is included in the financial disclosure and the revised cover letter.

Issue #4: We note that you have indicated that data from this study are available upon request. PLOS only allows data to be available upon request if there are legal or ethical restrictions on sharing data publicly. For information on unacceptable data access restrictions, please see http://journals.plos.org/plosone/s/data-availability#loc-unacceptable-data-access-restrictions.

Response: As you have advised the anonymized data set from this study will be available as a supporting information containing the minimum level of detail necessary to reproduce all numbers reported in the manuscript.

 

Response to Reviewer #1

Dear Daniel Milgrom MD,

Thank you very much for the review. Your constructive comments revealed some issues what we were unable to see. With your advices, we have revised the manuscript and we hope it will meet your expectations. If there is any further question or suggestion, we are happy to receive it.

Major issue #1: All ERCPs were done at outside institutions, and so only compares PTD to failed ERCPs. To claim PTD is essential, this would require a comparison to successful ERCP as well.

Response: 

Thank you for the comment. We agree on your opinion. Our tertiary referral center is a surgical unit where operations and interventional radiological procedures are routinely done. Our unit did not perform ERCP in the investigated period. As a limitation of this retrospective data collection, full data of the ERCPs performed in other units were not available. To narrow the limitations, we softened our conclusions to advocate less PTD over ERCP, and put work into collect more data about previously published literature regarding outcomes after ERCP and PTD results to get a better view from the differences or similarities. (manuscript: p15, line 240-313) 

As interventional gastroenterology has been added to our unit recently, we are planning a prospective study comparing ERCP and PTD in the near future.

Major issue #2: For Figure 1: From the text it is not clear regarding the timing of the of cholangitis. Is this saying that there is a higher rate of cholangitis after the ERCP, but before the PTD? Or is the cholangitis before both the ERCP and PTD? If you are saying that the cholangitis is after the ERCP, but before the PTD, it is important to state how many of the patients also had cholangitis prior to the ERCP. 

Response: 

Thank you for the comment. We agreed on that it was not clear from the text, thus we tried to explain it with more details.

In the hilar ERCP+PTD group 39 cholangitis cases were observed before the percutaneous drainage (between the ERCP and PTD), and 25 after the percutaneous drainage. The endoscopic procedures were done in other hospitals, and during the retrospective data collection data availability was limited about the number of patients suffering from cholangitis before the ERCP.

On the other hand, we consider it important that the obstruction which sustained the cholangitis was not solved by the ERCP and other intervention was urgently needed to save the patient regardless the cholangitis was the consequence of the primary disease or the complication of the failed ERCP. Better patient selection could improve the rate of failed ERCP-s, indicate adequate primary PTD and solve the obstruction and the cholangitis by itself, or with further conservative treatment.

We have included an updated Figure 1 in order to highlight the above mentioned comments, and made some clarification in the text (p11, line 182-183).

Major issue #3: I think your conclusions are overstated. You did not compare PTD to ERCP, but compared it to itself and failed ERCP. I think at most you can say that it is safe and effective and is appropriate for first choice in the treatment algorithm. Please consider revising this as well as the title of the manuscript. 

Response: Thank you for the advice. We have changed the conclusion and the title in order to avoid biased point of view and to be more objective about the facts. 

• Changed Conclusion:

 The results and especially the excellent success rates demonstrate that PTD is safe and effective, and it is appropriate for first choice in the treatment algorithm of perihilar stenosis. Ultimately, we concluded that PTD should be performed in experienced centers to achieve low mortality, morbidity, and high success rates.

• Changed Title:

 Percutaneous transhepatic drainage is safe and effective in biliary obstruction – A single-center experience of 599 patients

Minor issue #1: The formatting of Table 1 can be improved specifically where there are subcategories or lab values. 

Response: Thank you for the remark, we have corrected and improved Table 1, with highlighted subcategories.(p 10, line 163-165)

Minor issue #2: Table 3, please include p values for the totals that are mentioned in the text. 

Response: Thank you for the comment, we have included the p values in Table 3. (p12, line 188-190)

Minor issue #3: Please explain more about the complication rate per year as compared to the rate of internal drainage experience curve. 

Response: Thank you for the comment. 

We analyzed the total number of complications through the 12 years. A new figure (Fig. 2B) was added, and the original Fig. 2 was renamed as Fig. 2A. A higher peak point has been found during the early years and a decreasing complication rate over the years. This peak corresponds to the early courage of the less experienced interventionists which decreased with the rising experience (Fig. 2B). Although this does not follow precisely the learning curve on the Fig2A but, a major decrease can be observed in the number of complications, approximately between year 2011- 2012 (X-axis) where the internal-external / external ratio stabilized above 1. (p15, line 205-220)

Response to Reviewer #2

Dear Reviewer #2, 

We were happy to read your review and in-depth analysis which revealed some question that were left open in the original manuscript. You made clear that some explanations were not complete, and some of our conclusions were inaccurate or biased. We hope that our responses, explanations and the work we put into the revision will meet your expectations.

Major issue #1: Discussion – the authors advocate for PTD over ERCP; however, the current study does not compare PTD to ERCP. 

Additionally, the discussion fails to discuss/compare complications after PTD in the context of previously published literature regarding outcomes after ERCP in the setting of biliary obstruction. 

The discussion broadly approaches this but fails to discuss/compare the two in any granularity. As written, the current manuscript highlights that PTD can be performed safely in biliary obstruction with experienced clinicians.

Major issue #2: Discussion/Conclusion – The authors conclude that complications rates after PTD are comparable to ERCP and subsequently the authors advocate for PTD as the first choice for perihilar biliary obstruction. (1) The authors study does not compare complications rates between PTD and ERCP. (2) The discussion does not discuss complication rates/types/severity after ERCP in the context of previously published literature. As written, how can the authors make these conclusions?

Response for issues #1 and #2:

Thank you for helping us to see clear the overall picture. We agree on your opinion. Our tertiary referral center is a surgical unit where operations and interventional radiological procedures are routinely done. We did not perform ERCP during the investigated time period. As a limitation of this retrospective data collection, full data of the ERCPs performed in other units were not available. To narrow the limitations, we put more effort into search previously published literature regarding outcomes after ERCP and PTD results to get a better view from the differences or similarities between result of the mentioned interventions. (manuscript: p16, line 236-301) 

 On the other hand, we moderated our conclusions and suggestions to advocate less PTD over ERCP. We hope our softer conclusions will be less biased. (p16, line 240-313)

As interventional gastroenterology has been added to our unit recently, we are planning a prospective study comparing ERCP and PTD in the near future.

Major issue #3: Results – What was the technical success rate among the subgroup of patients that underwent PTD for perihilar biliary obstruction? This appears to be the key point of the manuscript, yet the technical success rate among this subgroup is not reported. As the authors point out, this represents a more technically challenging procedure.

Response:

Thank you for the relevant question. The technical success rate in the perihilar obstruction subgroup is 96.3% (13/357) which is similarly good as the overall technical success. This result has been also included in the revised manuscript. (p 9, line 137)

Major issue #4: Title – “Percutaneous transhepatic drainage is essential in perihilar biliary obstruction” is a strong statement. The title should be revised to avoid “overselling” the study, as this is a retrospective observational report and not a randomized controlled trial (this study does not compare primary PTD to primary ERCP). Additionally, the number of patients listed in the title is not actually the number of patients with perihilar biliary obstruction and should be revised or changed (357 patients had perihilar biliary obstruction).

Response: We agree on your advice. The changed the title is:

 “Percutaneous transhepatic drainage is safe and effective in biliary obstruction – A single-center experience of 599 patients”

Major issue #5: Results – the authors report 63 events where drain dislodgement occurred – these were classified as Clavien-Dindo Grade I or II complications (“minor” by the authors definition). Did none of these patients required a repeat intervention to have the drains replaced? By the authors definition, repeat intervention would be considered a “major” complication (Clavien-Dindo Grade III complications). Can the authors comment on how these 63 patients were managed if they did not require drain replacement? In a previous subsection, the authors state that 16 patients required reintervention in a 30-day period – were these counted as Clavien-Dindo III+ complications? It does not appear they were.

Response: Thank you very much for revealing this issue.

We understand and appreciate your comment. As a result we revised and reconsidered all drain dislocations and reinterventions. Dislocated (or obliterated) drains which needed reintervention were included in the major complication section. 

As a final result we have found 63 dislocations (including 2 obliterations): 39 early and 24 late dislocations. From the 39 early cases we have found 15 cases that were managed with radiological reinterventions (13 dislocation 2 obliteration)

In the 24 cases of patients who did not receive reintervention after the PTD dislocation there were different disease courses that did not made the reintervention possible or did not indicate a reintervention. Such case scenarios were: reposition was managed without true radiological intervention, internal-external drain dislocated to external position, resolution of cholangitis after the first PTD, disease progression or other organ failure. 

Thus in total we performed 16 reinterventions, due to dislocation or obliteration (n=15), or haemobilia (n=1). 

We modified the overall results and the result of the perihilar subgroup (including Table 2 and 3) analysis in accordance to your remark. The corrected Table 3 also includes the correction with cholangitis numbers as you suggested in your response #7. (manuscript: p9, line 150-190)

Major issue #6: Discussion – Should technical success rate be the primary metric in determining approach to biliary drainage for biliary obstruction? Several other factors play an important role in this decision making (i.e., therapeutic success rate, quality of life, durability, etiology/pathology, etc). Can the authors comment on this in their discussion?

Response: Thank you for your suggestion, you have raised an important point. We believe that the primary goal in a biliary obstruction is the resolution of the obstruction and the adequate biliary drainage. Considering that in the majority of the cases PTD happens in an advanced state of the disease, or as a salvage therapy when clinical success is hard to interpret. Not to mention the heterogenous definition of clinical success which makes the comparison even more complicated. That is why we highlighted technical success in our manuscript. Of course there are several other factors that play an important role in decision making not to mention patient preference. We worked on this section, and mentioned your suggestions with more detail in the revised discussion. (manuscript: p18, line 257-277)

Major issue #7: Results – cholangitis was not included as a “minor” complication after PTD for perihilar biliary obstruction – what is the rationale for excluding this complication, as it clearly his clinical implications?

Response: Thank you for the suggestion, we agree on your comment, it should have been included in the minor complications after PTD for perihilar biliary obstructions. We have modified the Table 3 and the result section as well. (p16, line 252-270)

Minor issue #1: Abbreviations (manuscript) – All abbreviations should be introduced and written out at first mention, followed by the abbreviated form for the rest of the manuscript.

Response: Thank you for your advice. The following abbreviations were introduced in the revised manuscript: Percutaneous transhepatic cholangiography (PTC), endoscopic retrograde cholangiopancreatography (ERCP,) computer tomography, magnetic resonance imaging.

Minor issue #2: . Table 1 – what does the “*” denote?

Response: It was mistakenly deleted: * no further localization is defined. It has been added to the Table 1 description in the revised manuscript.

Minor issue #3: Results – The first paragraph discussing one mortality should be included in the “complications” subsection and not as the first paragraph of the results section.

Response: We accept the suggestion, it has been included in the complication subsection.

Minor issue #4: Results (page 9, line 140-141) – The definition of Clavien-Dindo “minor” complication does not need repeating in the results section, as it is defined in the methods.

Response: We accept your remark, the repeated Clavien-Dindo minor complication definition has been deleted in the mentioned paragraph.

Minor issue #5: Paragraph 1 (a two-sentence paragraph) should be combined with paragraph 2

Response: Thank you for your advice. We combined the two paragraphs.

Minor issue #6: Discussion – throughout the manuscript the authors use the term “percutaneous transhepatic drain” but change to “percutaneous biliary catheter” in the discussion – terminology should remain consistent throughout the manuscript.

Response: Thank you, we agree on your comment. We have replaced the term [catheter] throughout the paper with [drain] to be more consistent.

Minor issue #7: Discussion (paragraph 2) – the authors introduce new data from their study in the discussion section (median serum bilirubin concentration) – all data should be included in the results section of the manuscript.

Response: Thank you for the remark. Median serum bilirubin concentration has been included in Table 1, results section.

Minor issue #8: Results – Procedural mortality was 1/599 – the authors report this as 0.001% - this is actually 0.17%.

Response: Thank you for the feedback, it has been corrected to 0.17%.

Again, thank you for giving us the opportunity to strengthen our manuscript with your valuable comments and queries. Since all the corrections have been made, we hope the manuscript will now be accepted without any further changes. We have worked hard to incorporate your feedback and hope that these revisions persuade you to accept our submission.

We look forward to hearing from you regarding our submission. We would be glad to respond to any further questions and comments that you may have.

Sincerely,

Ákos Szücs, MD PhD

Corresponding Author 

associate professor

deputy head of department, 

1st Department of Surgery and Interventional Gastroenterology 

Semmelweis University

e-mail: szucs.akos@gmail.com

tel: +36-20-8258916

---

## [Editor Report · Decision Letter 1]

5 Nov 2021

Percutaneous transhepatic drainage is safe and effective in biliary obstruction - A single-center experience of 599 patients

PONE-D-21-17286R1

Dear Dr. Szücs,

We’re pleased to inform you that your manuscript has been judged scientifically suitable for publication and will be formally accepted for publication once it meets all outstanding technical requirements.

Kind regards,

Leonidas G Koniaris, MD

Academic Editor

PLOS ONE
---

## [Editor Report · Acceptance letter]

9 Nov 2021

PONE-D-21-17286R1 

Percutaneous transhepatic drainage is safe and effective in biliary obstruction - A single-center experience of 599 patients 

Dear Dr. Szücs:

I'm pleased to inform you that your manuscript has been deemed suitable for publication in PLOS ONE. Congratulations! Your manuscript is now with our production department. 

Kind regards, 

on behalf of

Dr. Leonidas G Koniaris 

Academic Editor

PLOS ONE